# $^{233}$U/$^{236}$U signature allows to distinguish environmental emissions of civil nuclear industry from weapons fallout

K. Hain [1]✉, P. Steier [1], M.B. Froehlich [2], R. Golser[1], X. Hou [3], J. Lachner [1,7], T. Nomura[4], J. Qiao[3], F. Quinto [5] & A. Sakaguchi[6]

Isotopic ratios of radioactive releases into the environment are useful signatures for contamination source assessment. Uranium is known to behave conservatively in sea water so that a ratio of uranium trace isotopes may serve as a superior oceanographic tracer. Here we present data on the atomic $^{233}$U/$^{236}$U ratio analyzed in representative environmental samples finding ratios of $(0.1–3.7)\cdot10^{-2}$. The ratios detected in compartments of the environment affected by releases of nuclear power production or by weapons fallout differ by one order of magnitude. Significant amounts of $^{233}$U were only released in nuclear weapons fallout, either produced by fast neutron reactions or directly by $^{233}$U-fueled devices. This makes the $^{233}$U/$^{236}$U ratio a promising new fingerprint for radioactive emissions. Our findings indicate a higher release of $^{233}$U by nuclear weapons tests before the maximum of global fallout in 1963, setting constraints on the design of the nuclear weapons employed.

[1] Faculty of Physics, Isotope Physics, University of Vienna, Währinger Str. 17, 1090 Vienna, Austria. [2] Department of Nuclear Physics, Australian National University, Canberra ACT 2601, Australia. [3] Department of Environmental Engineering, Technical University of Denmark, DTU Risø Campus, 4000 Roskilde, Denmark. [4] Graduate School of Science, Hiroshima University, 1-3-1 Kagamiyama, Higashi-Hiroshima 739-8526, Japan. [5] Karlsruhe Institute of Technology (KIT), Institute for Nuclear Waste Disposal (INE), Hermann-von-Helmholtz Platz 1, 76344 Eggenstein-Leopoldshafen, Germany. [6] Center for Research in Isotopes and Environmental Dynamics, University of Tsukuba, 1-1-1 Tennodai, Tsukuba, Ibaraki 305-8577, Japan. [7] Present address: Helmholtz-Zentrum Dresden-Rossendorf, Bautzner Landstraße 400, 01328 Dresden, Germany. ✉email: karin.hain@univie.ac.at

The long-lived uranium (U) isotope $^{236}$U ($T_{1/2}$ = $2.342 \cdot 10^7$ years[1]) is increasingly adopted as environmental tracer with advantageous chemical properties especially for oceanography[2–5]. U shows a conservative behavior in sea water and therefore can be transported over long distances in this environment. In oxidizing conditions of surface and ocean water uranium is present in oxidation state +VI as uranyl ion ($UO_2^{2+}$)[6,7]. In the presence of complexing ligands such as carbonates or phosphates, which are readily available in ocean water, the solubility of U(VI) is enhanced by orders of magnitude[8–10] leading to the conservative behavior. An advantage of $^{236}$U compared with the naturally occurring and therefore more abundant U isotopes $^{234}$U, $^{235}$U, and $^{238}$U, is the high sensitivity to small anthropogenic U inputs to the large reservoir of environmental U. Considerable amounts of anthropogenic $^{236}$U have been released into the environment by atmospheric nuclear weapons testings, reprocessing plants and reactor accidents. The total deposition as global fallout has been estimated at 900–1400 kg[5,11] and from reprocessing plants at 115–250 kg[5]. This clearly dominates over the natural global inventory of this isotope (a few kg)[12]. For time-resolved archives, elevated $^{236}$U/$^{238}$U atom ratios can be attributed to a specific contamination source if the emission history is well-known. This has been successfully demonstrated for global fallout in coral cores from the Caribbean Sea[13] and the Northwest Pacific Ocean[4], for releases from the Sellafield reprocessing plant in sediments and water from the Irish Sea[14,15] and also for the contamination from the reactor accident in soils collected close to Chernobyl[16]. In these cases, the $^{236}$U/$^{238}$U atom ratio serves as tracer for ocean currents[5,17,18]. However, in systems affected by several contamination sources with complex water circulation processes, such as the Mediterranean Sea[19] or the Arctic Ocean[20], the lack of a second anthropogenic U isotope is a disadvantage compared with plutonium (Pu). Isotopic ratios of two anthropogenic nuclides, such as $^{240}$Pu/$^{239}$Pu, allow further insight than that provided by a single isotope concentration or its ratio to an also naturally occurring isotope like $^{238}$U. Isotopic ratios strongly depend on the production mechanism and hence, the emission source, which allows discrimination between possible origins of the contamination[21,22]. However, because Pu is a particle-reactive element, it shows high sedimentation rates in environmental waters (sea, river and fresh water reservoirs) because of the presence of colloids, and it is not well suited as a tracer for water transport processes. Recent technical developments at the Vienna Environmental Research Accelerator (VERA) enable low-level Accelerator Mass Spectrometry (AMS) measurements of anthropogenic $^{233}$U ($T_{1/2}$ = $1.592 \cdot 10^5$ years[1]) in the environment. The low environmental abundance of $^{233}$U is a challenge even for the highly sensitive detection technique AMS so that previous publications on $^{233}$U either analyzed its concentration in samples collected close to a contamination source[23,24] or used $^{233}$U, which had been added to the environmental sample, to normalize the $^{236}$U results for mass spectrometric measurements. The combination of environmental $^{233}$U and $^{236}$U could play a similar role for the U isotopic system as $^{240}$Pu/$^{239}$Pu for the Pu system. The isotopic ratio $^{233}$U/$^{236}$U stays undisturbed by chemical fractionation in the environment as well as during sample preparation, simplifying the interpretation of the measurement results in the presence of mixing and dilution processes.

In the present study, the abundances of both $^{233}$U and $^{236}$U and the resulting $^{233}$U/$^{236}$U atom ratios were investigated in samples from different natural reservoirs including corals from the Pacific Ocean, samples from the Irish and the Baltic Sea and peat bog samples from the Black Forest, Southwest Germany, which are influenced by different contamination sources. Our findings indicate a considerably lower production of $^{233}$U relative to $^{236}$U in thermal nuclear power plants compared with nuclear weapons

which agrees with our understanding of the possible production mechanisms of the two isotopes. Consequently, the $^{233}$U/$^{236}$U ratio has a great potential for emission source identification and hence as tracer for water transport processes. Furthermore, a time shift of −7 years in the deposition peak of $^{233}$U from global fallout compared with $^{236}$U was observed which indicates a more intensive use of enriched uranium in thermonuclear weapons or releases from $^{233}$U-fueled weapons during the early phase of the US nuclear weapons testings and gives insights into nuclear weapons design where details still remain classified.

## Results

**Sources for anthropogenic $^{233}$U.** In general, the main emission sources for anthropogenic radionuclides are either atmospheric nuclear weapons tests or nuclear industry, i.e., reprocessing plants or reactor accidents. Since the vast majority of nuclear power plants which have been in operation until today have used a thermal neutron spectrum and U as fuel, the production of $^{233}$U in nuclear reactors is strongly suppressed compared with $^{236}$U[25]. Both, official sources, e.g.[26–28] and unauthorized web sites[29] on nuclear weapons design are naturally scarce or impossible to verify. Yet, even though there are information sources stating that at least one nuclear weapons test using a mixture of $^{233}$U and $^{239}$Pu as fuel has been conducted ("Teapot MET", April 1955[29]), to our best knowledge, all nuclear weapon programs were clearly dominated by $^{235}$U or $^{238}$U, $^{239}$Pu based weapons[30]. In short, the most relevant production path for $^{233}$U via the reaction $^{235}$U(n,3n)$^{233}$U requires fast neutrons with energies above 13 MeV[31]. A contribution from the thorium fuel cycle[32] producing $^{233}$U by thermal neutron capture on $^{232}$Th can be considered as negligible. In contrast, $^{236}$U can be also produced in nuclear power plants and fission bombs via $^{235}$U (n,$\gamma$)$^{236}$U using thermal neutrons, apart from the production by fast neutrons in thermonuclear weapons via the reaction $^{238}$U (n,3n)$^{236}$U. Therefore, a significant production can be expected in thermonuclear weapons containing uranium enriched in $^{235}$U (sometimes referred to as oralloy). Fallout from the low-yield device "Teapot MET" (22 kt) mentioned in[29] can be assumed to be mainly locally restricted to the surrounding area of Nevada test-site (NTS)[33,34]. However, it is generally accepted that surface detonations of kilotons bombs cause tropospheric fallout, which is deposited in a band around the globe at the latitude of the test site (20°–50° N for NTS)[35]. Therefore, a contribution from the MET test to the total inventory of $^{233}$U at the latitude band of NTS is, in principle, possible but can be expected to decrease in an eastward direction[36]. A detailed discussion of the production mechanisms of $^{233}$U and $^{236}$U can be found in the Methods section.

**Selection of sample materials.** The $^{233}$U and $^{236}$U content of samples from five different locations, which are summarized in Table 1, were analyzed in this study. Samples comprises sea water and sediment as well as a peat and coral core. In four cases, chemically separated U in an iron oxide matrix was available from archived AMS sputter targets (the kind of sample holder suitable for the AMS ion source) in which $^{236}$U was previously determined. $^{233}$U is often added as a chemical yield tracer, however, only samples which have not been spiked with $^{233}$U during sample preparation were considered in the present work. If available, the $^{236}$U/$^{238}$U data obtained in the corresponding previous study are listed in Table 1 with one sigma uncertainty. A more detailed sample description can be found in the Methods section.

**Table 1 Overview of the sample material used in the present study for $^{233}U/^{236}U$ analysis.**

| Sample description | Number of samples | Coordinates of sampling station | Sampling year | Time range covered | Predominant contamination source | $^{236}U/^{238}U$ previous work | References |
|---|---|---|---|---|---|---|---|
| Irish Sea sediment core | 7 (3–47) cm depth | 54.416°N, 3.563°W | 1993 | – | Reprocessing plant Sellafield | $(1.35 - 4.36)\cdot10^{-5}$ | Srncik et al., Steier et al.[14, 56] |
| Irish Sea water IAEA-381(443) | 1 | 54.415°N, 3.560°W - 54.387°N, 3.558°W | 1993 | – | Reprocessing plant Sellafield | $(2.47 \pm 0.19)\cdot10^{-6}$ $(2.04 \pm 0.02)\cdot10^{-6}$ | Povinec et al., Pham et al.[15,57] Eigl et al.[3] |
| Peat core, Blackforest Germany | 23 | 48.718°N, 8.459°E | 2006 | 1921–1992 | Global fallout | $(0.53 \pm 0.09)\cdot10^{-6}$ to $(7.40 \pm 0.44)\cdot10^{-6}$ | Quinto et al.[37,38] |
| Coral core, (Kume Island) Pacific Ocean Japan | 31 | 26.319°N, 126.766 °E | 2014 | 1939–1970 annual resolution | Global fallout, tropospheric close-in fallout (PPG) by Castle (1954) & Hardtack I (1958) | $(0.09 \pm 0.01)\cdot10^{-9}$ to $(11.0 \pm 1.2)\cdot10^{-9}$ | Nomura et al.[40] |
| Baltic Sea water, Kattegat, Denmark | 2 | 56.57°N, 12.12°E 56.93°N, 12.20°E | 2015 | – | Global fallout, reprocessing plants La Hague & Sellafield, Chernobyl accident | – | – |

*PPG* Pacific Proving Grounds.

**The global fallout signature of $^{233}U/^{236}U$ in a peat bog core.** The $^{236}U/^{238}U$ and the $^{233}U/^{238}U$ atomic ratios measured in the peat core are plotted against the age of the respective peat layer in Fig. 1a, dated by using the unsupported $^{210}Pb$ method[37]. Values for the $^{236}U/^{238}U$ ratios range from $7.2 \cdot 10^{-7}$ to $9.2 \cdot 10^{-6}$ and for $^{233}U/^{238}U$ from $1.1 \cdot 10^{-8}$ to $1.8 \cdot 10^{-7}$ (see also Supplementary Table 1). The observed $^{233}U$ concentration in this environment, which is not directly influenced by any nuclear source except global fallout as shown by analyzing the Pu nuclide vector[38], is almost two orders of magnitude lower than the $^{236}U$ concentration. The $^{236}U/^{238}U$ data obtained in the present study agree reasonably well with the previously published data for the peat core[37] (compare Supplementary Fig. 1).

Both depth profiles of $^{236}U/^{238}U$ and $^{233}U/^{238}U$ ratios (Fig. 1a) show a pronounced peak with the maximum value in 1961.5 and in 1955.3, respectively. The explosion yield of atmospheric nuclear weapons tests is narrowly distributed with an expectation value of 1959.5 and a standard deviation of 3.1 years (see Fig. 1b). This means around 90% of the total explosion yield of all atmospheric weapons tests (440 Mt[39]) was released within only one decade and marks the most active phase of atmospheric nuclear weapons testing. Two main phases of atmospheric testing can be identified in Fig. 1b, i.e., 1952–1958 and 1961–1962, leading to the maximum global fallout in 1963, to which the $^{236}U/^{238}U$ maximum in the peat core was attributed[37,39]. The $^{236}U$ as well as the $^{233}U$ bomb peak detected in the peat core is approximated by Gaussian fits (black and gray solid lines) with similar widths, i.e., $19 \pm 1$ years and $18.6 \pm 0.9$ years (FWHM). As both nuclides were deposited predominantly during a rather narrow time interval, the peak shape results from migration of U in the peat. In contrast to the $^{233}U$ peak, the baseline of the $^{236}U/^{238}U$ does not reach pre-nuclear levels for younger layers as additional releases might have occurred. The resulting $^{233}U$ peak (peak center at AD $1953.5 \pm 0.5$) is shifted by 6.8 years towards older ages with respect to the $^{236}U$ bomb peak (peak center at AD $1960.3 \pm 0.4$).

This indicates that the maximum release of $^{233}U$ happened before the maximum deposition of global fallout and hence, can be attributed to the earlier testing phase, i.e., 1952–1958. Regarding the number of tests, the respective estimated yield and the altitude at which the tests were conducted, it can be deduced that atmospheric fallout from the earlier period was dominated by the U.S. program whereas the fallout maximum in 1963 was dominated by the USSR weapon tests[39] (see Supplementary Table 2). Considering the sampling location, the detected $^{233}U$ contamination can be therefore attributed either

to some early thermonuclear explosions conducted by the US at the Pacific Proving Grounds (PPG) which are said to have used oralloy as tamper material[29] or unfissioned $^{233}U$ from the "Teapot MET" explosion in 1955.

The overall $^{233}U/^{236}U$ ratio for nuclear weapons fallout was calculated from the peak area of the two Gaussian fits of the $^{236}U/^{238}U$ and the $^{233}U/^{238}U$ data. In both cases, the sample with the age AD 1920.7 and a $^{236}U/^{238}U$ and $^{233}U/^{238}U$ atom ratio of $(7.5 \pm 1.5) \cdot 10^{-7}$ and $(1.1 \pm 0.3) \cdot 10^{-9}$, respectively, serves as upper limit for the blank level which does not significantly affect the value of the overall $^{233}U/^{236}U$ isotopic ratio.

Dividing the peak area yields an average $^{233}U/^{236}U$ ratio of $(1.40 \pm 0.15)\cdot10^{-2}$. This value can be considered representative for compartments of the environment which do not preserve a high time resolution, and are only affected by global fallout. If the $^{236}U/^{238}U$ peak is disentangled according to the two phases of nuclear weapons testings (dashed black curves in Fig. 1a), a $^{233}U/^{236}U$ ratio of $(5.1 \pm 1.1)\cdot10^{-2}$ for the earlier phase is obtained (see Discussion section for details).

**The close-in fallout signature of the PPG in a coral core.** The $^{233}U/^{238}U$ and $^{236}U/^{238}U$ atom ratios determined in the corals from Kume Island are presented in Fig. 2 as a function of the age of the respective coral band. The corals were cut along layers with low density which correspond to the fast growth rates during summer time. Hence, the year 1951, e.g., refers to the time period from summer 1950 to summer 1951. The individual $^{236}U/^{238}U$ atom ratios are in very good agreement with the results from the previous study[40] (see also Supplementary Table 3). The stated $1\sigma$ uncertainties of the $^{233}U/^{236}U$ ratio are clearly dominated by the comparably low statistics in case of the $^{233}U$ measurement due to the low abundance of $^{233}U$ in the corals and the availability of material left in some AMS sputter targets from the previous study.

In general, the $^{236}U/^{238}U$ and the $^{233}U/^{238}U$ atom ratio with a maximum of $(1.05 \pm 0.05)\cdot10^{-8}$ and $(1.6 \pm 0.2)\cdot10^{-10}$, respectively, are almost three orders of magnitude lower than in the peat core. In the ocean water, fallout U is mixed with higher concentrations of natural U than in the peat bog so that the fallout signature is diluted before the U is concentrated in the corals. The level of the $^{236}U/^{238}U$ ratio for pre-nuclear samples is $(1.0 \pm 0.2)\cdot10^{-10}$ and $<3.1\cdot10^{-12}$ for $^{233}U/^{238}U$, respectively. Whereas two peaks of the $^{236}U/^{238}U$ data in 1954 and 1958 can be clearly identified in Fig. 2, there is only one maximum in the $^{233}U/^{238}U$ measurement data which is statistically significant, that is in the year 1958. The uncertainty of the $^{233}U/^{238}U$ ratio

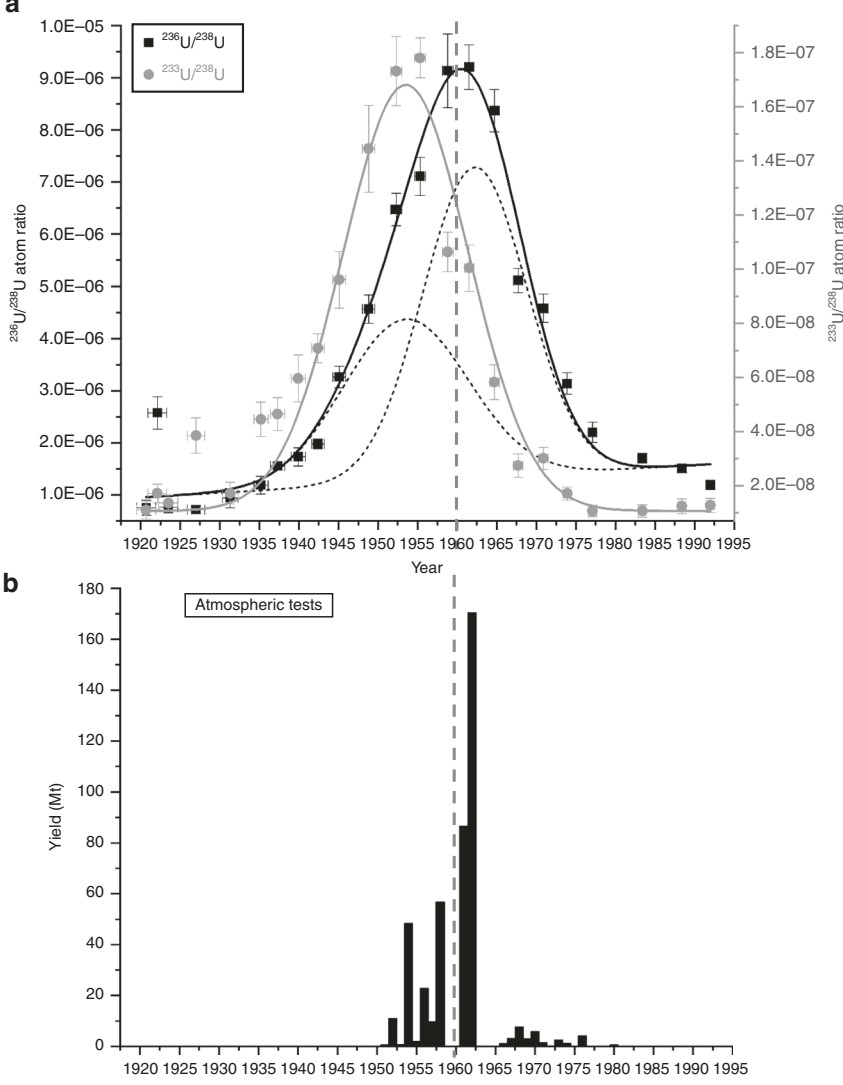

**Fig. 1** $^{233}$**U and** $^{236}$**U signal in the peat bog compared with weapons yields. a** Measured $^{236}$U/$^{238}$U (black squares) and $^{233}$U/$^{238}$U (gray dots) atom ratio in the peat core depending on the age of the peat layer. The measurement uncertainties shown are ±1 $\sigma$ (s.e.m.). The solid lines represent a Gaussian fit of the $^{236}$U (black line) and the $^{233}$U (gray line) data indicating a time shift between the $^{233}$U and the $^{236}$U release. The $^{236}$U peak can be disentangled by two Gaussian distributions (dashed black curves in **a**) corresponding to the two major testing phases of nuclear weapons with respect to the explosion yield (1952–1958 and 1961–1962) shown in **b**[39]. The vertical dashed gray line marks the year 1960 for a direct comparison of the time scales in **a** and **b**.

at 1955 unfortunately is too large to consider this data point as reliable. On the basis of the present data a maximum of the $^{233}$U/$^{238}$U atom ratio in 1955, therefore, cannot be unequivocally identified. The center of the maximum of the $^{233}$U/$^{238}$U ratio in 1958 coincides exactly with the maximum of the $^{236}$U/$^{238}$U ratio which shows that also the $^{233}$U/$^{238}$U ratio in the corals is strongly affected by the close-in fallout from the PPG. Following the argumentation given by Nomura et al.[40] who attributed the second peak at 1958 to the operation Hardtack I, our results suggest a considerable use of oralloy during this test series. However, no information about the tamper material in operation Hardtack is available to us at present. While no good data was obtained for the year 1955, there is clearly no maximum in 1954 corresponding to the first peak in the $^{236}$U/$^{238}$U atom ratio. This finding indicates that large quantities of $^{236}$U, but not of $^{233}$U, have been produced by the devices tested before 1954. This is in good agreement with the claim that Castle Nectar in 1954 was the first thermonuclear explosion

with an oralloy tamper[29]. It also agrees with the assumption that the Ivy King test in 1952 was a pure oralloy fission device[41] and hence, did not generate enough fast neutrons required for the build-up of $^{233}$U. Nevertheless, the $^{233}$U abundance in the marine environment of the Pacific Ocean seems to gradually increase from 1953 on, suggesting that $^{233}$U has been produced from the very first thermonuclear weapons, even though to a much smaller extent.

The weighted average of the $^{233}$U/$^{236}$U ratio (see Fig. 3) was calculated from the measured $^{233}$U/$^{238}$U and $^{236}$U/$^{238}$U ratios for three time periods (I–III) that are characterized by a different $^{233}$U/$^{236}$U ratio. The ratios for samples before 1949 are not shown in this figure, as in most cases only upper limits for the $^{233}$U/$^{236}$U ratio were obtained because of the low $^{233}$U concentrations. In period I with no significant $^{233}$U production, i.e., until 1956, the average $^{233}$U/$^{236}$U = $(0.31 \pm 0.07) \cdot 10^{-2}$ is much lower than for the period 1957–1962. Period II is characterized by an increased release of $^{233}$U probably caused

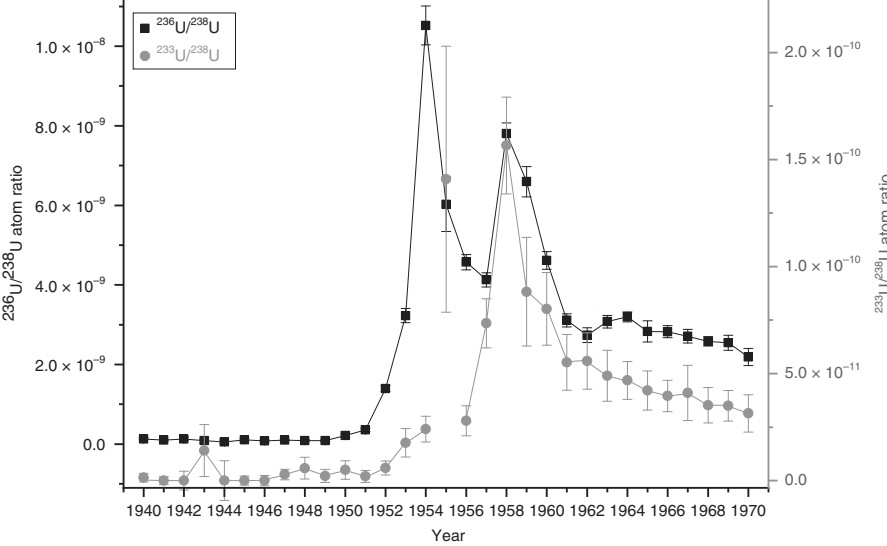

**Fig. 2 Measurement results from coral samples.** $^{236}U/^{238}U$ (black squares) and $^{233}U/^{238}U$ atom ratio (gray dots) in the Kume coral core as a function of time. The measurement uncertainties shown are ±1 $\sigma$ (s.e.m.). The data points are connected by solid lines (bold for $^{236}U$) to guide the eye. The maximum at 1955 is not statistically significant and thus, not linked to the neighboring points.

by the close-in fallout of operation Hardtack I and an average $^{233}U/^{236}U$ ratio of $(1.81 \pm 0.15) \cdot 10^{-2}$ was obtained. The "1950" sample and also the "1955" sample, which corresponds to a possible first maximum in Fig. 2, show an elevated ratio but do not considerably affect the weighted average due to the large uncertainties. Starting from the year 1963 coinciding with the maximum of global fallout (period III), the ratio levels out to an average of $(1.44 \pm 0.12) \cdot 10^{-2}$ which is consistent with the global fallout average determined from the Western Europe peat samples.

**The signature of nuclear power production in the Irish Sea.** The $^{233}U/^{238}U$ ratios detected in Irish Sea sediment range from $9.6 \cdot 10^{-9}$ to $5.9 \cdot 10^{-8}$ and, hence, are comparable to the ratios found in the peat core (see also Supplementary Table 4 for details). The $^{233}U/^{236}U$ ratios of three samples, which have been diluted by a factor of 100, show a high uncertainty (Fig. 4) and therefore have a low significance for the interpretation of the $^{233}U/^{236}U$ ratios in the sediment.

The $^{233}U$ count rate from the undiluted samples was four orders of magnitude higher than from a U sample considered as instrumental blank for $^{233}U$. Consequently, a clear $^{233}U$ signal was detected, but as shown by the depth profile in Fig. 4, the $^{233}U/^{236}U$ ratios in the sediment core are significantly lower than in the peat and the coral core. The weighted average from the sediment samples ($n = 7$) results in $^{233}U/^{236}U = (0.13 \pm 0.02) \cdot 10^{-2}$, which is consistent with the ratio determined in Irish Sea water of $(0.11 \pm 0.01) \cdot 10^{-2}$. Hence, the weighted average of $^{233}U/^{236}U = (0.12 \pm 0.01) \cdot 10^{-2}$ in the Irish Sea, close to the reprocessing plant Sellafield, is one order of magnitude lower than in nuclear weapons fallout found in the peat and coral core. In accordance with the theoretical discussion of the $^{233}U$ and $^{236}U$ production mechanisms in the Methods section, we attribute this low ratio to the U releases from the reprocessing plant because it indicates the lack of neutrons with energies above the threshold for the $^{235}U$ (n,3n)$^{233}U$ reaction. The elevated ratio of the sample from 19 cm depth deviates significantly from the calculated average; nevertheless it also clearly shows the low ratio expected for reactor dominated anthropogenic input.

**Mixing of different source terms in the Danish straits.** The measured $^{233}U/^{238}U$, $^{236}U/^{238}U$ and $^{233}U/^{236}U$ ratios in two samples from the Danish straits (Kattegat) are given in Table 2. These two samples were collected in a similar region at a distance of only ~40 km from each other and, as expected, show very similar values for the three atom ratios. The $^{236}U/^{238}U$ ratio is clearly elevated with respect to the natural abundance, which confirms the mainly anthropogenic origin of $^{236}U$ in the Danish straits. The $^{233}U/^{238}U$ ratio is quite low and comparable to the ratios found in the modern layers of the coral core from the Pacific Ocean. Within the uncertainties the $^{233}U/^{236}U$ ratios of the two samples are indistinguishable and the resulting average of $(0.45 \pm 0.02) \cdot 10^{-2}$ is situated between the value attributed to the reprocessing plant Sellafield $(0.12 \pm 0.01) \cdot 10^{-2}$ and to the global fallout $(1.40 \pm 0.15) \cdot 10^{-2}$. This is consistent with the picture of the Danish straits being a mixing zone of water masses carrying global fallout signature with waters containing uranium originating from the reprocessing plants as well as fallout from the Chernobyl accident[42,43].

As discussed in the previous section, no difference in the $^{233}U/^{236}U$ ratio between reprocessing plants and NPPs can be expected, and in this way no differentiation between a Chernobyl and a La Hague/Sellafield fraction is possible. However, the contribution of uranium from generic nuclear fuel and global fallout can be calculated by using a two end member linear mixing model, as commonly applied to Pu ratios, e.g., in[21]. The average $^{233}U/^{236}U$ ratio of the two Kattegat water samples and the $^{233}U/^{236}U$ ratio of global fallout $(1.40 \pm 0.15) \cdot 10^{-2}$ from the peat core and that of nuclear fuel $(0.12 \pm 0.01) \cdot 10^{-2}$ from the Irish Sea sediments yields a global fallout fraction of $(25.8 \pm 3.4)\%$ at the sampling location in the Danish straits. As expected, the larger contribution comes from the nuclear power industry which is most probably caused by the considerable releases from the reprocessing plants as discussed before in this paper and in previous publications[5,18,42].

## Discussion

Environments affected by the Sellafield reprocessing plant or by nuclear weapons fallout were found to differ by one order of

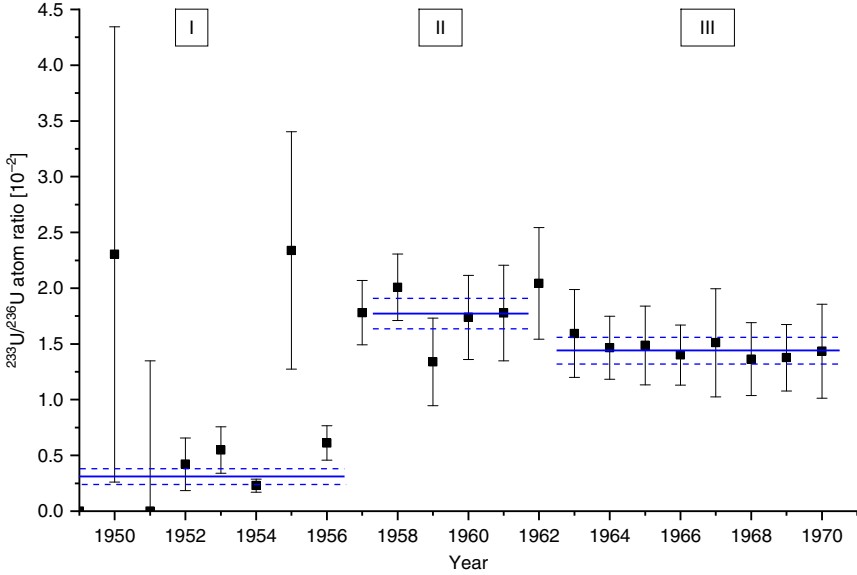

**Fig. 3** $^{233}$**U/**$^{236}$**U ratio in coral samples.** $^{233}$U/$^{236}$U (black squares) calculated from the measurement results for the Kume coral core with ±1 $\sigma$ uncertainty. The solid blue lines indicate the weighted average for the respective time period (I–III) and the dashed lines the corresponding 1 $\sigma$ uncertainty (s.e.m.).

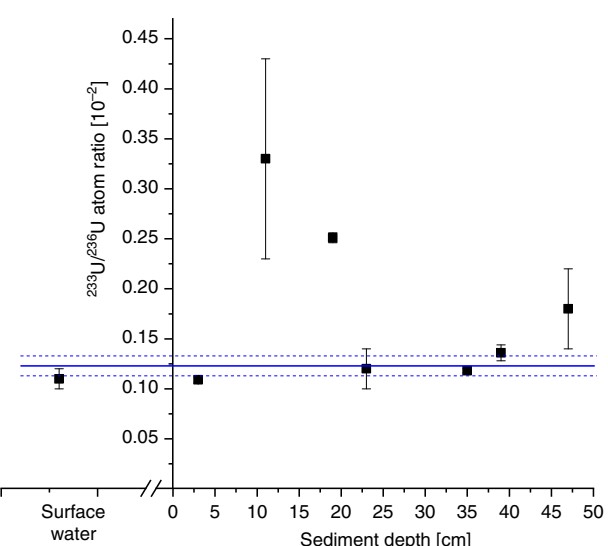

**Fig. 4** $^{233}$**U/**$^{236}$**U ratio in the Irish Sea.** Depth profile of the $^{233}$U/$^{236}$U atom ratio in the Irish Sea sediment core collected close to the Sellafield reprocessing plant and in Irish Sea water (IAEA-381) with ±1 $\sigma$ uncertainty. Increased uncertainties of three samples at depths 11 cm, 23 cm and 47 cm are caused by low counting statistics on $^{233}$U due to preceding dilution (1:100) of the material. The horizontal blue line marks the weighted average for the $^{233}$U/$^{236}$U ratio in the Irish Sea with ±1 $\sigma$ uncertainty (s.e.m.).

magnitude in their $^{233}$U/$^{236}$U atom ratio. Reservoirs exposed to global fallout from nuclear weapons testings showed a $^{233}$U/$^{236}$U ratio of around $1.4 \cdot 10^{-2}$. Depending on the contribution from close-in (PPG) fallout $^{233}$U/$^{236}$U ratios up to $1.8 \cdot 10^{-2}$ were found in a coral core from the Pacific Ocean. In contrast, the very low $^{233}$U/$^{236}$U ratio of $(0.12 \pm 0.01) \cdot 10^{-2}$ detected close to Sellafield can be generally assigned to spent fuel from thermal NPPs. By analyzing the $^{240}$Pu/$^{239}$Pu atom ratio and the $^{238}$Pu/$^{239+240}$Pu activity ratio, it has been shown that the present sediment core is neither influenced by the reprocessing of weapon-grade plutonium from the early operation of the reprocessing plant nor by

global fallout, but carries the signature of high burn-up Pu which is characteristic for spent fuel from NPPs[14]. At present, the large majority of all NPPs in operation is still based on the U fuel cycle and uses a thermal neutron spectrum[44], so that the different $^{233}$U/$^{236}$U ratio allows a discrimination between emissions from civil nuclear industry and nuclear weapons fallout.

This finding is in good agreement with our theoretical considerations on the production of $^{233}$U (see "Methods" section for details) from which we conclude that a significant production of $^{233}$U is only possible in thermonuclear weapons using a tamper of highly enriched $^{235}$U but not in U-based thermal nuclear power plants. However, the $^{233}$U/$^{236}$U ratio from the period of testing at the PPG (II) is too low to explain the ratio found for global fallout, as the peat samples suggest a considerable dilution of the $^{233}$U signal from the PPG by $^{236}$U produced in a later phase of atmospheric nuclear weapons testing. If we assume that the input of anthropogenic uranium (U) can be attributed in a simplified view to only two time intervals corresponding to the two most active phase of atmospheric nuclear weapons testings, i.e., 1952–1958 and 1961/62 (compare Fig. 1b), then we can disentangle the isotopic signatures of the two phases. The dashed black curves in Fig. 1a show the fit of the corresponding Gaussian peaks to the data from the peat core. For this, we assume that all 233U originates from the PPG and thus shows up in the peak corresponding to the earlier phase of testing. This fixes the time offset of the peat data vs. the UNSCEAR data[39] for the explosion yields (Fig. 1b) to −2.3 years and the width of the distribution with $\sigma = 16.8$ years. The offset is calculated from a shift of the $^{233}$U peak center at 1953.5 compared with the center of the distribution published by UNSCEAR at 1955.8 and is probably caused by the migration of U in the peat. The time difference between the $^{233}$U/$^{238}$U and the $^{236}$U/$^{238}$U peak center is taken as 5.9 years according to the UNSCEAR data. The ratio of the areas for the earlier time interval corresponds to $^{233}$U/$^{236}$U = $(5.1 \pm 1.1) \cdot 10^{-2}$. This is significantly higher than source ratio of $(1.81 \pm 0.15) \cdot 10^{-2}$ determined for the PPG in the coral samples. This points to an additional source of $^{233}$U for which the $^{233}$U-fueled "Teapot MET" explosion is a potential candidate. In order to estimate the contribution from the "Teapot MET" test to the global fallout, the source function of $^{233}$U could be assessed sampling at different distances to the Nevada Test Site. As we can explain the peat data

**Table 2 $^{236}$U/$^{238}$U, $^{233}$U/$^{238}$U and $^{233}$U/$^{236}$U results for two selected samples collected at the strait between Denmark and Sweden.**

| Sample name | Location | $^{233}$U counts | $^{233}$U/$^{238}$U | $^{236}$U/$^{238}$U | $^{233}$U/$^{236}$U [$10^{-2}$] |
|---|---|---|---|---|---|
| 2015-0587 | 56.57° N, 12.12°E | 396 | $(6.08 \pm 0.31) \cdot 10^{-11}$ | $(1.34 \pm 0.02) \cdot 10^{-8}$ | $0.45 \pm 0.02$ |
| 2015-0622 | 56.93°N, 12.20°E | 513 | $(5.15 \pm 0.47) \cdot 10^{-11}$ | $(1.18 \pm 0.03) \cdot 10^{-8}$ | $0.44 \pm 0.04$ |

Uncertainties are given in $\pm 1 \sigma$ (s.e.m.).

without a $^{233}$U source in the later testing phase (1961–1962), a substantial use of oralloy in thermonuclear weapons seems unlikely. This phase was dominated by the USSR testing for which little information is available. At least, initial plans of the USSR suggest the use of natural or even depleted uranium as tamper in the RDS-37 device[45], which was the first staged thermonuclear bomb tested by the USSR in November 1955[29]. Our findings, therefore, impose constraints on the weapons design and the resulting source terms also for other radionuclides released into the environment by nuclear weapons tests.

For close-in fallout from the PPG, our data from the coral core already suggest a strong correlation between the $^{233}$U/$^{236}$U ratio and the $^{235}$U enrichment of the thermonuclear device. One of the reported aims of Hardtack I was the development of tactical nuclear weapons which should result in smaller and more efficient devices[26]. As it is known that a higher yield per mass of the device and thus a higher efficiency can be achieved by replacing natural uranium tamper by oralloy[46], it is well conceivable that such devices were tested during Hardtack I. In addition, elevated $^{236}$U/$^{239}$Pu ratios found in corals at Enewetak Atoll from the same time period also point to an oralloy tamper in one or more of the explosions during this test series[47,48]. Analysis of local fallout $^{233}$U/$^{236}$U ratio from test sites can, therefore, add information on the fuel used during the different test series. Samples from the former USSR test sites Semipalatinsk (Kazakhstan) and Novaya Zemlya (Arctic Ocean) would be especially interesting because even less information on the USSR weapons design is available.

In addition, the study of the $^{233}$U/$^{236}$U ratios at the different test sites is necessary to assess the effect of in situ production of $^{233}$U by neutron capture on $^{232}$Th contained in the soil/rocks which might offer the possibility to distinguish also local and global fallout. This could be especially relevant for surface and low altitude explosions, during which the surface material was exposed to the nuclear fire ball and got incorporated into the blast of the explosion[49]. Child and Hotchkis[24] found $^{233}$U concentrations up to a factor of 100 higher within a 200 km zone around ground zero at Emu field and on the Montebello Islands[24] which they attributed to in situ production, followed by mobilization of the irradiated material. In contrast to this spatially very restricted local fallout from low-yield weapons tests[39], global fallout is known to have been caused by the large, in particular thermonuclear tests conducted at the PPG by the US and at Novaya Zemlya test site by the USSR[50]. Therefore, small scale explosions, conducted also at other test sites like the Semipalatinsk or Nevada test site, can be considered as a negligible contribution to the in situ produced $^{233}$U in global fallout. The estimation of the global budget of in situ produced $^{233}$U is very difficult, as the production and mobilization rate depends on a number of parameters which are not sufficiently well known to the authors. The $^{233}$U data from the peat core indicate a minor contribution from the Novaya Zemlya test site where the test program was dominated by air tests[39] in contrast to the near-surface explosions at the PPG where coral sand from the atolls was incorporated into the blast. For example, the Tsar bomb in October 1961 (50 Mt explosion yield) is known to

have strongly affected the surface of Novaya Zemlya[51,52] so that a mobilization of irradiated material and thus in situ produced $^{233}$U, could be expected. Nevertheless, the measured $^{233}$U data do not show a significant contribution from the later testing phase in 1961–62.

Apart from the mentioned local effects, the large potential for emission source identification in combination with the conservative behavior of U in oxic natural waters, makes the $^{233}$U/$^{236}$U ratio well suited for tracing environmental transport processes. Compared with other mobile radionuclides, which are already used for oceanography, e.g., $^{137}$Cs, $^{129}$I, $^{99}$Tc and the corresponding ratios, the $^{233}$U/$^{236}$U ratio is independent from the emission history of the specific source, as it only depends on the fuel and the neutron spectrum and can be considered as more reliable. Especially, in complex oceanographic settings like the Baltic Sea with several contamination sources, i.e., global fallout, the reprocessing plants La Hague and Sellafield and fallout from Chernobyl, the detection of $^{233}$U/$^{238}$U in addition to $^{236}$U/$^{238}$U can quantify the contribution from global fallout. This was demonstrated for two samples collected at the Danish straits (Kattegat) for which a global fallout contribution of 25% was obtained. In case of a nuclear accident, the surplus of potential U releases to the background due to nuclear weapons can now be quantified.

In addition to the samples analyzed in the present study, which were collected in the Northern Hemisphere, future studies should aim to map the $^{233}$U/$^{236}$U ratio for the Southern Hemisphere. As observed for Pu isotopes, isotopic ratios on the Southern Hemisphere show a higher variability[53–55], because weapons fallout is dominated by the local fallout of the French tests in French Polynesia and the British tests in Australia. A next step in method development will be establishing a standard material for $^{233}$U/$^{236}$U/$^{238}$U to be shared with other AMS laboratories.

## Methods

**Detailed sample description and preparation.** The $^{233}$U/$^{236}$U atom ratio was determined in seven samples from different depths of a sediment core (3–47 cm) from the Irish Sea which was collected by the Federal Maritime and Hydrographic Agency, Germany. Previous work suggests that the core showed no good stratigraphy[14,56] but was probably mixed by environmental reworking. The $^{236}$U/$^{238}$U ratios determined in the sediment samples ranged from $1.35 \cdot 10^{-5}$ to $4.36 \cdot 10^{-5}$ and thus, were considerably elevated compared with natural background ($10^{-14}$–$10^{-10}$[12]) or global fallout ($\approx 5 \cdot 10^{-8}$[11]). The samples were clearly affected by the large amounts of uranium from spent fuel discharged by the Sellafield reprocessing plant. At the beginning of the plant operation in the 1950s and 60s, effluents were characterized by a low $^{240}$Pu/$^{239}$Pu isotopic ratio (<0.07) due to the production of weapon-grade Pu. Later emissions originated from the reprocessing of spent fuel from thermal Nuclear Power Plants showed $^{240}$Pu/$^{239}$Pu ratios higher than 0.20[21]. Dating of the sediment core using its $^{241}$Pu content resulted in a maximum age of $34.0 \pm 0.4$ years before 2010[56] and, therefore, an isotopic signature of high burn-up fuel from NPPs was expected. This has been confirmed by the high $^{240}$Pu/$^{239}$Pu ratios ranging from 0.20 to 0.33 detected in the sediment core[14]. Since the $^{236}$U count rates detected from the Irish Sea sediment were considerably larger than 1000 s$^{-1}$ in[14], and thus prone to cross-contamination, the first batch of three original samples was diluted by approximately a factor 100.

In addition to the sediment samples, Irish Sea water, i.e., the certified reference material IAEA-381, now available as IAEA-443[57], was analyzed with respect to its $^{233}$U/$^{236}$U atom ratio within the scope of the present study. For this sample, no

archived separated U target material was available and thus it was prepared from the original reference material.

A time-resolved profile of the $^{236}$U concentration over the past 80 years starting from 1992 as youngest age in an undisturbed ombrotrophic peat core was obtained in[37]. The layers of the peat core were dated by using the unsupported $^{210}$Pb method. The data for the $^{236}$U/$^{238}$U ratio showed a clear bomb peak with a maximum $^{236}$U/$^{238}$U ratio of (7.4 ± 0.4)·10$^{-6}$ at a depth corresponding to AD 1959. This is in good agreement with the most active phase of atmospheric nuclear weapons testing in the late 1950s and early 1960s. The analysis of the Pu isotopic ratios supports the finding that the peat is exclusively affected by global fallout[38]. Within the scope of the present study, the $^{233}$U and $^{236}$U concentrations were analyzed in 23 samples from the peat core covering the relevant time span from 1921 to 1992.

Due to their annual growth bands, corals represent a high-resolution archive of U, which is incorporated into the carbonate skeleton of the corals from the ocean water.

The $^{236}$U/$^{238}$U ratios detected in the corals from Kume Island were clearly influenced by the close-in fallout[50] from the high-yield nuclear weapons tests conducted by the USA at the Pacific Proving Grounds (PPG), the Marshall Islands. The PPG are located in the North Equatorial current, which transported the close-in fallout west and then, turning into the Kuroshio current, to the northeast, passing the location of Kume Island. Two distinct peaks were found with a maximum $^{236}$U/$^{238}$U ratio of (11.0 ± 1.2)·10$^{-9}$ and (8.55 ± 1.17)·10$^{-9}$, respectively, in the years of most intense weapons testing at the PPG regarding the yield of the explosions (Supplementary Table 1). Consequently, the two maxima were attributed to the two largest test series at the PPG, i.e., Operation Castle in 1954 and Hardtack I in 1958. In the present study, the $^{233}$U/$^{236}$U ratio was analyzed in coral samples from Kume Island corresponding to the time interval from 1939 to 1970 with annual resolution.

From an on-going project on the distribution and temporal evolution of the $^{236}$U concentration in the Baltic Sea[42], two samples collected in the strait between Denmark and Sweden (Kattegat) were chosen for additional $^{233}$U analysis. The Danish straits are a very interesting maritime environment for the study of the physio-chemical behavior of U, as brackish water from the Baltic Sea mixes with saline water from the North Sea and the Atlantic Ocean. Using the distribution of $^{129}$I and $^{233}$U in the North Sea, it has been shown[2,18] that releases from Sellafield and in particular from La Hague, are transported towards the Danish coast by the respective sea currents. This leads to the mixing of uranium originating from the reprocessing plants with U from global fallout. Accordingly, $^{236}$U/$^{238}$U ratios detected in sea water from the Danish straits were found to be around four times higher than expected from global fallout indicating the presence of at least one additional contamination source[42]. Here, emissions from the reprocessing plants as well as fallout from the Chernobyl accident have to be considered as possible explanation.

Accelerator Mass Spectrometry requires the uranium to be embedded in several mg of solid material, usually Fe$_2$O$_3$, which means that a specific sample preparation has to be applied to the original environmental sample. A detailed description of the chemical extraction and purification of uranium from the different types of sample material is given in the respective previous publications, i.e., Irish Sea sediment[14], peat bog[37], corals from Kume Island[40] and Baltic Sea water[58]. Some archived uranium samples pressed into sputter targets for the ion source of VERA could be directly re-used for the present work. A slightly modified procedure for the actinide separation[59] was applied to 200 mL of the reference material IAEA-381 Irish Sea water. U was first pre-concentrated using a Fe(OH)$_3$ co-precipitation and then purified using the UTEVA© resin from Eichrom Technologies, Inc. The final process step of the AMS sample preparation in all projects was a co-precipitation of U with Fe(OH)$_3$ which was then converted to Fe$_2$O$_3$ by calcination in a furnace at 800–900° C. From the Irish Sea sediment different aliquots were prepared in the initial project with variations in the sample preparation. For the present work, we selected aliquots without any spike and thus, in some cases used different aliquots from the same original sample material than those published in ref. [14]. For the dilution of three of the samples, the sample material was removed from the respective archived sputter target and dissolved in 8.5 M HCl. A total of 100 mg of additional Fe were dissolved in each sample solution. Then, U was co-precipitated with Fe(OH)$_3$ using NH$_4$OH. Starting from the Fe(OH)$_3$, AMS targets were prepared as described before. As the $^{233}$U count rates from the diluted samples were rather small (around 10$^{-3}$ s$^{-1}$), a second set of sediment samples were measured in the original sample holder without additional treatment.

**AMS measurement procedure**. $^{236}$U and $^{233}$U were detected by the ultra-sensitive detection method Accelerator Mass Spectrometry (AMS) at the Vienna Environmental Research Accelerator (VERA), Austria. Actinides, in particular $^{236}$U, are routinely analyzed at the AMS facility VERA and the corresponding set-up and measurement procedure has been described in detail before, and most recently in[60]. In short, U is extracted from a cesium sputter ion source as UO$^-$ molecules and then injected into the accelerator, operated at a terminal voltage of 1.65 MV for actinide measurements. A relatively high helium pressure is used in the terminal stripper to suppress the molecular background[61], resulting in a stripping yield of around 21% for the selected charge state 3$^+$. Isotopic background on the high-energy side, mainly due to $^{235}$U and $^{232}$Th, which is injected into the accelerator as

$^{235}$U$^{16}$O$^1$H$^-$ and $^{231}$Th$^{16}$O$^1$H$^-$, is efficiently filtered out by a 90° analyzing magnet, a Wien filter, an electrostatic analyzer, and a second, recently installed, 90° magnet. The remaining ions are identified in the final detection system, a Bragg type ionization chamber[62]. With this setup an overall detection efficiency of 5·10$^{-4}$ and a detection limit for $^{236}$U/U below 10$^{-11}$ was achieved[63]. The detection efficiencies for $^{233}$U and $^{236}$U can be assumed to be comparable due to the similar mass of the two isotopes. For a uranium oxide target prepared from our in-house $^{236}$U standard Vienna-KkU ($^{236}$U/$^{238}$U = (6.98 ± 0.32)·10$^{-11}$[12]) mixed with Fe$_2$O$_3$ (1:30), which can be considered as machine blank for $^{233}$U, an upper limit for the $^{233}$U/U ratio of 6·10$^{-13}$ was measured. However, possible contamination from sample preparation are not included in this blank. Therefore, we made the conservative assumption that our blank level is equal or below the concentration of the sample with the lowest count rate of $^{233}$U and $^{236}$U events, respectively, for each sample material.

The $^{236}$U and $^{233}$U count rates in the ionization chamber were normalized to the $^{238}$U current measured in a Faraday cup after the second electrostatic analyzer in order to take into account fluctuations in the source output and the ion optical transmission through the accelerator. The difference in detection efficiency between the Faraday cup and the ionization chamber was corrected by using the in-house standard Vienna-KkU. A dead time correction was applied to the detector count rates of the undiluted samples from the Irish Sea sediment.

## Comparison of $^{233}$U and $^{236}$U production

The thermal neutron capture on $^{235}$U and in particular the $^{238}$U (n,3n)$^{236}$U reaction induced by fast neutrons in thermonuclear explosions have been previously identified as the most important production channels for $^{236}$U[11,12]. For the build-up of $^{233}$U, the following nuclear reactions and decay processes have to be considered:

$$^{235}\text{U}(n, 3n)^{233}\text{U} \tag{1}$$

$$^{238}\text{U}(n, 2n)^{237}\text{U} \rightarrow {}^{237}\text{Np} \rightarrow {}^{233}\text{Pa} \rightarrow {}^{233}\text{U} \tag{2}$$

$$^{234}\text{U}(n, 2n)^{233}\text{U} \tag{3}$$

$$^{232}\text{Th}(n, \gamma)^{233}\text{Th} \rightarrow {}^{233}\text{Pa} \rightarrow {}^{233}\text{U} \tag{4}$$

The (n, 2n) and (n, 3n) reactions in (1), (2) and (3) require threshold energies of 6 MeV[64] and 13 MeV[31], respectively. A production via these reactions in nature, therefore, is only possible by neutrons from cosmic rays at the presence of U, which is limited to the shallow subsurface of the Earth's crust (upper 2 m). In addition, the cross-sections for the (n,3n) and (n,2n) reactions on $^{235}$U, $^{238}$U, and $^{234}$U are low, i.e., below 1 barn, as demonstrated in Supplementary Fig. 2, showing F/EXFOR data[65,66]. Consequently, these reaction channels are negligible compared with thermal neutron capture on $^{232}$Th (reaction (4)), which has a cross-section of 7.37 barn, especially in minerals with elevated $^{232}$Th content, e.g., monazite. Peppard et al. demonstrated the presence of natural $^{233}$U in pitchblende and Brazilian monazite concentrate by isolating the decay product $^{225}$Ac. They observed a mass ratio of $^{233}$U/$^{238}$U of (1.3 ± 0.2)·10$^{-13}$ in pitchblende and (4 ± 2)·10$^{-11}$[67] in monazite. We have detected isotopic ratios of $^{233}$U/U in yellowcake samples in the range of several 10$^{-14}$ at the Vienna Environmental Research Accelerator (VERA), as part of this work.

When discussing anthropogenic sources for $^{233}$U, it has to be noted that from a neutronics point of view, $^{233}$U is very well suited as fuel for nuclear power plants and also for nuclear weapons. According to reaction (4), it can be efficiently bred from $^{232}$Th which is the starting point for the proposed thorium fuel cycle[32]. In that way, 1500 kg of $^{233}$U were synthesized in the USA[68]. Several reactor prototypes were operated especially in the 60s and 70s, e.g., the pebble-bed reactor in Germany, or the DRAGON experimental reactor in England. However, these prototypes or research reactors are by far outnumbered by the industrial application of the uranium fuel cycle in NPPs using a thermal neutron spectrum and the reprocessing of the fuel involved. Only extremely small amounts of $^{233}$U are produced in U-fueled NPPs because of the small cross-sections involved

(reactions (1)–(3)). Furthermore, the average energy of neutrons emitted by the thermal fission of $^{233}$U is around 2 MeV with the energy distribution for the number of neutrons $N(E)$ being described by the well-known Watt spectrum. Integration of the corresponding empirical formula

$$N(E) = 4.75 \cdot 10^6 \ \sinh{(2E)}^{0.5} \cdot e^{-E} \qquad (5)$$

with $E$ given in MeV, results in a fraction of only 2.4% of all fission neutrons with energies above the threshold for (n,2n) reactions, i.e., 6 MeV, and of 0.01% with energies above 13 MeV, the threshold for (n,3n) reactions. In contrast, $^{235}$U shows a thermal neutron capture cross-section of 95 b[1], so that considerable amounts of $^{236}$U are produced in NPPs. This difference in the production mechanisms of $^{233}$U and $^{236}$U is supported by the very low $^{233}$U/$^{236}$U ratios of $1 \cdot 10^{-6}$ obtained by reactor model calculations for the fuel of pressurized water reactors[25].

Apart from the direct release by $^{233}$U-fueled weapons, i.e., the "MET" explosion of operation "Teapot" mentioned before, the only relevant production path for $^{233}$U seems to be the reaction $^{235}$U(n,3n)$^{233}$U. Neutrons with energies of 14.1 MeV are provided in a thermonuclear device by the fusion reaction of deuterium and tritium[69] via

$$d + t \rightarrow \alpha \ (3.5 \text{ MeV}) + n (14.1 \text{ MeV}) \qquad (6)$$

Supplementary Table 1 gives an overview over the largest explosions regarding the yield during the era of atmospheric testing. The largest pure fission bomb tested purportedly using enriched $^{235}$U, Ivy King, had a yield of 0.5 Mt[41,70]. For weapons tests with higher explosion yields, a device-dependent percentage of the energy is produced by fusion reactions that means neutrons above the threshold energy for the build-up of $^{233}$U were released in the corresponding devices. Comparing Supplementary Fig. 1, the cross-section for the $^{235}$U(n,3n)$^{233}$U reaction at this neutron energy is maximum 0.1 b whereas the $^{238}$U(n,3n)$^{236}$U reaction has a cross-section of around 0.5 b. Therefore, a high $^{233}$U/$^{236}$U ratio can only be expected in thermonuclear weapons containing uranium enriched in $^{235}$U (e.g., oralloy, more than 90 % enriched[30]). It has been reported that in the well-known Teller-Ulam configuration of thermonuclear weapons, oralloy was used as so-called blanket or tamper in a few devices which exploded during the period of atmospheric testings, i.e., Castle Nectar (1954) and Redwing Cherokee (1956)[29] with an explosion yield of 1.7 Mt and 3.8 Mt, respectively.

Due to the limited experimental data on the production cross-section of $^{233}$U (compare Supplementary Fig. 1) and the lack of information on the construction details of nuclear devices, a theoretical prediction of the $^{233}$U/$^{236}$U ratio in global fallout can only be roughly estimated. Assuming a nuclear device with an average enrichment of 60% and only considering the 14 MeV cross-sections for the respective (n,3n) reactions results in a $^{233}$U/$^{236}$U ratio of 4 %. Production of $^{236}$U via neutron capture on $^{235}$U can be neglected as the corresponding cross-section is below $10^{-3}$ b for 14 MeV neutrons[65]. This theoretical value for the $^{233}$U/$^{236}$U ratio only serves as an estimate for the order of magnitude; the production of both nuclides is more complicated than described before, because of possible destruction and repeated capture processes. As most nuclear devices are supposed to have been equipped with a tamper made from natural uranium[29], the ratio in global fallout is probably further decreased. In general, the environmental $^{233}$U concentrations are expected to be about 100 times lower than those of $^{236}$U, i.e., $^{233}$U/$^{236}$U $\approx$ 1 %. To summarize, on average, fallout from nuclear weapons tests should show a higher $^{233}$U/$^{236}$U ratio than emissions from thermal nuclear power plants or reprocessing plants which is in agreement with our measurement results and allows source identification for environmental contamination.

## Data availability

The source data underlying Figs. 1a, 2, 3 and 4 and Table 2 are provided as Source Data files on https://vera2.rad.univie.ac.at/share/WWW_Exchange/public/Hain2020_Uranium-233/

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

## Acknowledgments

## Author contributions

K.H. wrote the manuscript and performed the AMS measurements together with P.S., who initiated the study. Data evaluation and interpretation was done by K.H., P.S., J.L. and R.G. M.B.F. was responsible for the sample preparation of the Irish Sea sediments, T.N. and A.S. for the coral core, F.Q. prepared the peat bog core, and J.Q. and X.H. provided and prepared the water samples from the Baltic Sea. X.H. added valuable information on nuclear weapons fallout.

## Competing interests

The authors declare no competing interests.
