## [Peer Review File · Nature Communications]

Reviewers' comments:

Reviewer #1 (Remarks to the Author):

Hain et al. $^{233}\text{U}/^{236}\text{U}$ review

This is novel work that will be of wide interest for those in the field of environmental radioactivity, nuclear forensics, the history of nuclear bomb tests, and the use of fissionogenic nuclides as environmental tracers in oceanographic and atmospheric transport/mixing. The title signals new data and approaches but would be better with an additional element related to the history of nuclear bomb tests to draw in the readers. I would expect to see other groups move towards validation of such methods and I can envisage this U ratio used as a sophisticated tracing tool in a number of settings.

Hain et al present a suite of ^{233}U analyses by AMS for a range of environmental archives – peat, coral, sediments and ocean water - to demonstrate the advantage of using the isotope pair $^{233}\text{U}/^{236}\text{U}$ over ^{236}U alone in nuclear forensics and environmental tracing. This study builds on work undertaken thus far by this and other groups for ^{236}U , indeed it looks again at many of the same samples prepared for earlier work. It also extends the work done by many groups using Pu-isotope pairs, but is much more useful because of the conservative behaviour of U in contrast to particle-reactive Pu.

Uranium-233 abundance is extremely low in all samples, and especially so for coralline material from the Pacific ($^{233}\text{U} < 3$ attomoles/g for peak bomb test levels). Analyses are thus challenging and some are conducted for material expected to have negligible concentrations of ^{233}U because they pre-date anthropogenic weapons testing.

This work serves to provide both proof-of-principle and also useful constraints on the characteristic fallout of early bomb tests, for which many details remain classified or lost. This work reveals the changing nature of fallout from earlier test series, with ^{235}U -enriched tamper source of ^{233}U or ^{233}U fuelled devices, to post 1961-2 tests, which produced much less ^{233}U and enhanced ^{236}U .

Comments/questions:

Please declare the ^{238}U concentration to allow estimation of the abundance of ^{233}U and ^{236}U . This may well have been determined for spiked analyses previously.

The explanation for the systematic difference for $^{236}\text{U}/^{238}\text{U}$ between this and earlier studies is not adequate. To state that improved normalisation methods have brought about a shift in estimates concentration begs the question 'What was wrong with earlier methods?'. Please respond. Why are the published $^{236}\text{U}/^{238}\text{U}$ results listed in Supplementary Tables. Why not use plot to illustrate systematic(?) differences?

What prospects are there for a widely-adopted $^{233}\text{U}/^{238}\text{U}$ standard? The use of an internal standard is OK for proof-of-principle, but eventually, different groups will need to compare results. Make suggestions within the text.

Use of Gaussian fits for range of test series with known duration – problematic

Deconvolving the mixture (2-component). What methods were used here?

Line435.. 'hardly provided' do you mean extremely rare, none at all.. please clarify.

line437, low abundance of ^{234}U (ie. $^{234}\text{U}/^{238}\text{U} \approx 5.47\text{e-}5$ in natural U) does not dictate the yield of $^{234}\text{U}(n,2n)^{233}\text{U}$.. it contributes, reader needs also to know cross section and threshold energies. This is revealed later in lines459-461 and Supp Info).

Ratios as percent – awkward. Not used for Pu isotopes. Imagine if you will a ratio of 1.01, albeit unlikely here: Would one quote a ratio of 101%.. or indeed 10.1% for 0.101? There will be settings (near prototype fast breeder reactors, for example) where one might see much higher $^{233}\text{U}/^{236}\text{U}$.

In Supp Info.. quote ratio as ($\times 10^{-2}$) rather than % .

The methods section is long and reproduces much of what is main text.

Throughout, I prefer oralloy not Orallooy.. Is this term widely used today? In line480 is $>90\%$ a critical threshold (no pun intended).. don't you just mean highly-enriched.

Statistical significance of the 'single' ^{233}U peak in the coral sample. Please declare test used to determine this.

David A. Richards, Bristol

Reviewer #2 (Remarks to the Author):

This manuscript presents novel and highly valuable results and in my opinion it definitely deserves to be published. The authors introduce a new tool to trace back the events that have occurred during the "nuclear era" by using the most modern technology based on accelerator mass spectrometry, which was developed in their lab during the last decade. In overall their findings are novel and impressive. Interestingly, even though the authors do not discuss in detail the production and potential releases of ^{233}U in the thorium fuel cycle, their conclusions regarding distinctive periods of " ^{233}U production" (see page 12) match very well with the history of thorium fuel cycle (to my knowledge the first publications on this subject appeared around 1956 to 1959, see e.g. MURBACH, EW; HANSEN, WN. PYROPROCESSING THORIUM FUELS. INDUSTRIAL AND ENGINEERING CHEMISTRY, V 51, 177-178, published in 1959).

In overall, the reported results look accurate and reliable. Some of the measurement results are associated with relatively large uncertainties, but I believe it does not diminish the value of this study as a whole. It is understandable that measurement of isotopes with extremely low abundances like those of ^{233}U and ^{236}U in environmental samples is technically very challenging. I hope that the authors will be able to further improve their analytical technology and I am looking forward to learn soon about even more sensitive analyses.

I recommend publishing this manuscript after minor revision with account to comments below.

Introduction:

For the completeness of the subject description I would suggest to mention the production of ^{236}U from ^{240}Pu decay as well as to make a reference to thorium fuel cycle when discussing ^{233}U production in introduction rather than at the end of the manuscript (Reference [62]).

It would be interesting to hear authors' opinion on U-236 that originates from industrial or military applications, for instance from the use of depleted uranium that might contain U-236.

Discussion

I would rather disagree with conclusion "that a significant production of ^{233}U is only possible in thermonuclear weapons" (page 16, line 268); please see my comment above about thorium fuel cycle as well as figures on page 22 of this manuscript (lines 445 – 450).

Because the section on pages 22 to 25 is entitled "Comparison of ^{233}U and ^{236}U production" I would suggest to also add nuclear reactions that lead to production of U-236. That would make a comparison easier.

Response to Referees

„ $^{233}\text{U}/^{236}\text{U}$ - a new tracer for environmental processes and nuclear forensics “

Ms. Ref. No.: NCOMMS-19-28566

Reviewer 1:

Comment 1:

Please declare the ^{238}U concentration to allow estimation of the abundance of ^{233}U and ^{236}U . This may well have been determined for spiked analyses previously.

We added the ^{238}U concentrations of the peat bog, the coral and the Irish Sea water samples to the respective tables in the Supporting Information (SI). ^{238}U concentrations in the Irish Sea sediment unfortunately have not been determined in the previous studies. The ^{238}U concentrations in the two samples from the Danish Straits are part of a larger data set which have been analyzed within the frame of a different study by our collaboration partner and will be published separately.

Comment 2:

The explanation for the systematic difference for $^{236}\text{U}/^{238}\text{U}$ between this and earlier studies is not adequate. To state that improved normalisation methods have brought about a shift in estimates concentration begs the question ‘What was wrong with earlier methods?’. Please respond. Why are the published $^{236}\text{U}/^{238}\text{U}$ results listed in Supplementary Tables. Why not use plot to illustrate systematic(?) differences?

We now show the comparison between the data published in Quinto et al, 2013 and the present work in the following which we have also included in the SI:

Indeed, the differences appear systematic which suggests a problem in the normalization. As said in SI of the present manuscript we now use external standard materials for normalization which were not available in 2013. As mentioned in Quinto et al, 2013, the $^{238}\text{U}5+$ currents were very low (table S2: typically around 0.2 pA) so that count rates of $^{234}\text{U}5+$ as well as $^{235}\text{U}5+$ were analyzed in addition. The isotopic ratios are essentially based on the ^{236}U and ^{234}U count rates assuming natural abundance of ^{234}U and equal detection efficiency for the two isotopes. As can be seen from table S2 in Quinto et al 2013, the number of detected ^{234}U events is rather small for normalization (82 – 916 detected events in total). When the measurements for Quinto 2013 were carried out, the intention was to normalize to the ^{235}U count rate. Since $^{235}\text{U}/^{238}\text{U}$ showed limited agreement with the natural abundance (around 0.56%, see table S2 Quinto et al, 2013), ^{234}U was chosen as isotope for normalization. For the present measurements we normalize the detector events to the $^{238}\text{U}3+$ current and the different efficiency for current and count rate measurements are corrected by the standard materials.

We have scrutinized the old data and could not identify a clear reason for the deviation. The efficiency for the two isotopes ^{236}U and ^{234}U in the previous measurement must have been different. With the present procedure such a deviation would be detected by the standard material and corrected for.

For the study of Quinto, 2013 we want to mention that the absolute $^{236}\text{U}/^{238}\text{U}$ ratio does to affect the interpretation.

Comment 3:

What prospects are there for a widely-adopted $^{233}\text{U}/^{238}\text{U}$ standard? The use of an internal standard is OK for proof-of-principle, but eventually, different groups will need to compare results. Make suggestions within the text.

We have added “A next step in method development will be establishing a standard material for $^{233}\text{U}/^{236}\text{U}/^{238}\text{U}$ to be shared by other AMS laboratories.” in the main text. We plan to spike

a substantial amount of our in-house standard Vienna-KkU with ^{233}U . We hope to arouse interest for ^{233}U at other AMS labs and win them for an intercomparison exercise.

Unfortunately, the next two lines in the reviewer comments are somewhat unclear and we have difficulties to refer it to a specific section in the manuscript. Anyhow, we try to give clarification.

Comment 4:

Use of Gaussian fits for range of test series with known duration – problematic

The duration of the test series (earlier phase: 6 years and later phase: 2 years) is that short that it will not have influenced the depth distribution of the corresponding signal in the peat which have a width (FWHM) of around 19 years. For the diffusion of the radionuclides in the peat, a Gaussian distribution seems to be the most reasonable model. We do not fully get the point of the reviewer's comment and think that our interpretation is valid.

Comment 5:

Deconvolving the mixture (2-component). What methods were used here?

We assume that this comment refers to the 2-end mixing model (line 246), while in fact there is no sophisticated mathematical background:

$$R_{\text{mixed}} = a \cdot R_{\text{GF}} + (1-a) R_{\text{NPP}}$$

GF: global fallout

NPP: nuclear power plant

with "a" being the unknown to be determined. A more detailed description can be found in the reference [21].

Comment 6+7:

Line435.. 'hardly provided' do you mean extremely rare, none at all.. please clarify.

line437, low abundance of ^{234}U (ie. $^{234}\text{U}/^{238}\text{U} \approx 5.47\text{e-}5$ in natural U) does not dictate the yield of $^{234}\text{U}(n,2n)^{233}\text{U}$.. it contributes, reader needs also to know cross section and threshold energies. This is revealed later in lines459-461 and Supp Info).

Neutrons with these high energies are only present in cosmic rays which are strongly attenuated with increasing depth from the Earth's surface.

In order to be more precise and to shorten the text (taking into account comment 10 of the referee) we have rewritten this section:

Line 434: "The (n,3n) and (n,2n) reactions in (1), (2) and (3) require threshold energies of 6MeV [61] and 13MeV [31], respectively. A production via these reactions in nature therefore is only possible by neutrons from cosmic rays at the presence of U, which is limited to the shallow subsurface of the Earth's crust (upper 2 m). In addition, the cross-sections for the (n,3n) and (n,2n) reactions on ^{235}U , ^{238}U , and ^{234}U are low, i.e. below 1barn, as demonstrated in Supplementary Figure 1, showing ENDF/EXFOR data [62,63].

Consequently, these reaction channels are negligible compared to thermal neutron capture on ^{232}Th (reaction (4)), which has a cross-section of 7.37barn, especially in minerals with elevated ^{232}Th content, e.g. monazite."

We deleted in line 459: “In addition, the cross-sections for the (n,3n) and (n,2n) reactions on ²³⁵U, ²³⁸U, and ²³⁴U are very low, i.e. below 1barn, as demonstrated in Supplementary Figure 1, showing ENDF/EXFOR data [62,63].”

Comment 8+9:

Ratios as percent – awkward. Not used for Pu isotopes. Imagine if you will a ratio of 1.01, albeit unlikely here: Would one quote a ratio of 101%.. or indeed 10.1% for 0.101? There will be settings (near prototype fast breeder reactors, for example) where one might see much higher ²³³U/²³⁶U.

In Supp Info.. quote ratio as (x 10⁻²) rather than % .

²³³U/²³⁶U ratios are now quoted as 10⁻² in the main manuscript as well as in the supplementary information.

Comment 10:

The methods section is long and reproduces much of what is main text.

As our manuscript is going to be the first publication on the application of environmental ²³³U for source identification, we think that the methods sections has to be rather detailed to demonstrate the reliability of our experimental procedure and theoretical interpretation of the measurement data. Due to the length restrictions of nature communications this could not be included directly in the main text body so that we have to repeat some of the assertions made there also methods sections. Nevertheless, we tried shorten the discussion of the ²³³U production (see comment 6+7).

We deleted in Line 468: “...which requires fast neutrons with energies above 13 MeV [31]”
Line 483 has been rewritten to: Due to the limited experimental data on the production cross-section of ²³³U (compare Supplementary Figure 1) and the lack of information on the construction details of nuclear devices, a theoretical prediction...”

Comment 11:

Throughout, I prefer oralloy not Oralloy.. Is this term widely used today? In line480 is >90% a critical threshold (no pun intended). don't you just mean highly-enriched.

According to our references the term “Oralloy” i.e. “Oak Ridge alloy” not just refers to highly enriched uranium (HEU) but to weapons grade uranium, i.e. which is enriched in ²³⁵U to more than 90% and therefore is relevant for the present manuscript. Consequently, and I dare say fortunately, the term “Oralloy” is not widely used nowadays, because HEU is already quite unusual for civil nuclear reactors. In fact, 90% enrichment is a critical threshold for U to be referred to as “Oralloy”, while U with lower enrichment is referred to as HEU. In our opinion, the term “oralloy” is more compact than “weapons grade uranium”.

We now use consistently “oralloy” with lower case only.

Comment 12:

Statistical significance of the ‘single’ ²³³U peak in the coral sample. Please declare test used to determine this.

The data points for the $^{233}\text{U}/^{238}\text{U}$ ratio from 1942 to 1952 were fitted by a horizontal line yielding a baseline of $(4.7 \pm 1.55) \cdot 10^{-12}$. Starting with 1952 an increasing trend can be observed. The uncertainty of the baseline was added to the uncertainty of the “1958” data point. The difference between the $^{233}\text{U}/^{238}\text{U}$ ratio of the “1958” data point and the baseline was divided by the sum uncertainty. We obtain the result that the “1958” data point is 6.2σ above the baseline and therefore, is statistically significant. A discovery is often said to be significant only with more than 5σ deviation from the baseline, though this is a rather conservative approach. In contrast, the “1955” data point has a larger uncertainty and is only 2.1σ above the baseline and therefore is considered as statistically not significant.

Reviewer 2:

Introduction:

Comment 1:

For the completeness of the subject description I would suggest to mention the production of U-236 from Pu-240 decay as well as to make a reference to thorium fuel cycle when discussing U-233 production in introduction rather than at the end of the manuscript (Reference [62]).

Any of the production channels are mentioned in the introduction. We therefore assume that the reviewer is referring to the first section of the “Results” section which covers the sources of anthropogenic ^{233}U .

We agree that the production of ^{236}U from ^{240}Pu decay in principle is possible but due to the long half-life of ^{240}Pu (~6500 yr) this production channel so far is not relevant for the environmental concentrations of ^{236}U . As our manuscript is already close to the maximum length for nature comm. we hope the referee understands that we only focus on the production and decay channels which are relevant for discussing our measurement results. Regarding the thorium cycle we have added in this section in line 86: “...and U as fuel,…” and in line 93: “A contribution from the thorium fuel cycle [32] producing ^{233}U by thermal neutron capture on ^{232}Th can be considered as negligible.”

Comment 2:

It would be interesting to hear authors’ opinion on U-236 that originates from industrial or military applications, for instance from the use of depleted uranium that might contain U-236.

This is a very interesting question, indeed. In fact, the $^{236}\text{U}/^{238}\text{U}$ ratio in depleted U is elevated (10^{-5} - 10^{-6}) compared to ratios measured in the environment which have been attributed to global fallout (10^{-7} - 10^{-9}). This ratio of depleted U is similar to the ratios sometimes measured in commercial reagents and artificial standards. The presence of depleted uranium in the environment like in ammunition is locally restricted and shows a low mobility because of the large particle size and the low solubility so that it can be expected to stay mainly in the region where it was used. If a small fraction does get dissolved and enters the ground water or riverine systems, the $^{236}\text{U}/^{238}\text{U}$ ratio is diluted quickly with the omnipresent natural ^{238}U when

it is transported onwards. We therefore consider the contribution of depleted uranium to the global inventory of ^{236}U as negligible compared to nuclear weapons fallout.

For further information, please refer to:

B. Salbu, et al. Oxidation states of uranium in depleted uranium particles from Kuwait, *J Environ Radioact*, 78 (2005) 125–135

G. Jia, et al. Concentration and characteristics of depleted uranium in biological and water samples collected in Bosnia and Herzegovina, *J Environ Radioact*, 89 (2006) 172-187

Discussion:

Comment 3:

I would rather disagree with conclusion “that a significant production of ^{233}U is only possible in thermonuclear weapons” (page 16, line 268); please see my comment above about thorium fuel cycle as well as figures on page 22 of this manuscript (lines 445 – 450).

The statement we made here in the first place was probably too general. To be more precise, we added in line 276: “... not in **U-based** thermal nuclear power plants.”

Comment 4:

Because the section on pages 22 to 25 is entitled “Comparison of ^{233}U and ^{236}U production” I would suggest to also add nuclear reactions that lead to production of U-236. That would make a comparison easier.

For a better comparison, we have added in line 429:

“The thermal neutron capture on ^{235}U and in particular the $^{238}\text{U}(n,3n)^{236}\text{U}$ reaction induced by fast neutrons in thermonuclear explosions have been previously identified as the most important production channels for ^{236}U [11,12].”

REVIEWERS' COMMENTS:

Reviewer #1 (Remarks to the Author):

I am pleased to declare that my comments have been addressed with appropriate rebuttal or amendment to the text/additional figures.

Reviewer #2 (Remarks to the Author):

In my opinion the points raised in the previous round of review have been satisfactorily addressed by the authors in the revised manuscript and I do not see any reasons for any further revision. I congratulate authors with this excellent study and recommend accepting the revised manuscript for publication in its current form.

Response to Referees

„²³³U/²³⁶U: a new tracer for environmental processes and nuclear forensics “

Ms. Ref. No.: NCOMMS-19-28566B

Reviewer 1: *I am pleased to declare that my comments have been addressed with appropriate rebuttal or amendment to the text/additional figures.*

Answer: No additional comments which need to be addressed.

Reviewer 2: *In my opinion the points raised in the previous round of review have been satisfactorily addressed by the authors in the revised manuscript and I do not see any reasons for any further revision. I congratulate authors with this excellent study and recommend accepting the revised manuscript for publication in its current form.*

Answer: No additional comments which need to be addressed.